# Generalized Behavior Learning from Diverse Demonstrations

**Varshith Sreeramdass, Rohan Paleja, Letian Chen,**
**Sanne van Waveren, Matthew Gombolay**
School of Interactive Computing, Georgia Institute of Technology
{vsreeramdass, rpaleja3, letian.chen, sanne}@gatech.edu,
matthew.gombolay@cc.gatech.edu

**Abstract:** Learning robot control policies through Reinforcement Learning can be challenging due to the complexity of designing rewards, which often result in unexpected behaviors. Imitation Learning overcomes this issue by using demonstrations to create policies that mimic expert behaviors. However, experts often demonstrate varied approaches to tasks. Capturing this variability is crucial for understanding and adapting to diverse scenarios. Prior methods capture variability by optimizing for behavior diversity alongside imitation. Yet, naive formulations of diversity can result in meaningless representation of latent factors, hindering generalization to novel scenarios. We propose Guided Strategy Discovery (GSD), a novel regularization method that specifically promotes expert-specified, task-relevant diversity. In the recovery of unseen expert behaviors, GSD improves 11% over the next best baseline across three continuous control tasks on average. Code is available online at https://github.com/CORE-Robotics-Lab/GSD.

## 1 Introduction

Autonomous robots are increasingly being deployed, for instance, in manufacturing, allowing for improved working conditions and production rates [1, 2, 3]. However, as many of these systems are expensive to program and unable to handle dynamic settings, fields of Reinforcement Learning (RL), Inverse RL (IRL), and Imitation Learning (IL) have attempted to learn control policies using alternative methods. RL [4, 5, 6, 7] requires tasks to be specified by rewards that can be difficult to design and are known to give rise to unexpected behaviors [8]. IL [9], instead, is a more accessible paradigm to human end-users, where agents learn policies by directly mimicking provided demonstrations. IRL [10] aims to infer rewards from demonstrations. However, human-provided demonstrations can significantly vary in their characteristics, even when a task is well-specified [11], due to latent, often continuous, preferences that different humans possess [12, 13, 14].

Capturing multimodality in human demonstrations can help identify latent preferences. These preferences can be leveraged towards personalized human-robot interaction [15] or collaboration [16, 17]. However, with continuous latent factors (e.g., different speeds at which a robot quadruped runs), it may not be feasible to obtain demonstrations that densely cover the latent factor space. It is thus important to construct a "generalizable latent space", which generalizes from limited demonstrations to effectively produce behaviors corresponding to novel latent variations (e.g., learning to run at two m/s from demonstrations with speeds of one m/s and three m/s). Generalization with unobserved factors can be difficult, as the dimensions along which demonstrations vary need to be accurately identified and individual demonstrations located, before extending to novel behaviors. Prior IL methods address capturing variations in demonstrations [18, 19] by employing a diversity objective formulated as mutual information (MI) between states and a learned latent variable. Works in unsupervised RL that focus on meaningfully structuring the latent space reveal that naively modeling MI can lead to behaviors that "overfit" to states [20] or are specific to certain objectives [21]. We show that such "overfitting," or specificity can be problematic for generalization in IL.

To learn a generalizable latent space, it is important to capture variations and structure in demonstrations without compromising task performance. We formulate such a novel form of diversity that

First Workshop on Out-of-Distribution Generalization in Robotics at CoRL 2023.

we term as *task-relevant diversity*. **Our key insight:** If task-relevant diversity could be specifically encouraged, generalization over latent factors would be better achieved.

We propose Guided Strategy Discovery (GSD), an IL method that uses task-relevance guided diversity regularization to learn novel behaviors. Our contributions are three-fold: 1) We show the need for reformulating diversity for generalization in multimodal IL and introduce task-relevant diversity, a form that retains task performance. 2) We propose GSD, a multimodal IL algorithm that uses derived task-relevance to guide diversity regularization. 3) We evaluate various methods for generalization through metrics measuring recovery of ground truth factors (known during evaluation) and show that GSD reduces error over baselines by 11% in two robot control and manipulation domains.

## 2 Problem Statement and Preliminaries

We consider an infinite-horizon, discounted Markov decision process (MDP), $(S, A, P, \rho_0, \gamma)$, where $S$ and $A$ represent state and action spaces, $P \colon S \times A \times S \to \mathbb{R}$, the transition probabilities, $\rho_0 \colon S \to \mathbb{R}$, the initial state distribution, $\gamma$, the discount factor. A set of demonstrations $\mathscr{D}$ is collected from an optimal expert policy $\pi^\xi$, governed by latent factors, i.e., $\mathscr{D}$ consists of $\tau_i^\xi = \{s_0, a_0, s_1, a_1, ...\}$, $a_t \sim \pi^\xi(\cdot|s_t, \omega_i)$, $s_{t+1} \sim P(\cdot|s_t, a_t)$, and $\omega_i$ the latent factor value. We aim to learn a policy $\pi$ that captures expert behavior within and outside the dataset $\mathscr{D}$ without access to $\omega_i$ or $\Omega$.

We consider IL from $\mathscr{D}$ in the online framework of Generative Adversarial Imitation Learning (GAIL) [22] for its data-sample efficiency. It learns the policy $\pi$ by minimizing the Jenson-Shannon divergence between stationary distributions induced by the policy and the expert $\pi^\xi$. Utilizing a discriminator $D$, to model the expert's distribution, optimization is set up as an adversarial two-player game between $\pi$ and $D$. The GAIL objective is $J^{\text{GAIL}} := E_\pi[\log D(s, a)] + E_{\pi^\xi}[\log(1 - D(s, a))]$.

We base our approach for multimodal IL on InfoGAIL [18, 19]. To capture multiple modalities, InfoGAIL extends GAIL by using a latent strategy variable $z \in Z$ with a prior $p_z(\cdot)$ to condition the policy $\pi(\cdot|s, z)$. It optimizes MI between the latent variable and state-action pairs $I(z; s, a)$ which is combined with $J^{\text{GAIL}}$ with an importance weight, $\lambda_I \in [0, 1]$. MI is optimized indirectly by a variational lower bound [23] where the intractable posterior, $p(z|s, a)$, is approximated using a learned function, $q(z|s, a)$. We refer to $q$ as the *strategy decoder* as it infers $z$ from the state action pair, $(s, a)$. The objective of InfoGAIL is $J^{\text{InfoGAIL}} := (1 - \lambda_I)J^{\text{GAIL}} + \lambda_I E_{z, \pi}[\log q(z|s, a)]$.

## 3 Need for Regularization

MI encourages diversity in behaviors by rewarding the visitation of states associated with distinct $z$s. In InfoGAIL, one may expect increased importance for diversity $\lambda_I$ to result in behaviors that generalize the demonstrations. Fig. 1b (bottom right) shows the effect of increasing $\lambda_I$ in PointMaze, a 2D domain with continuous state and action spaces (see B.1). Diversity in behaviors is either insufficient (several clump together) or arbitrary (no pattern that governs deviation from demonstrations).

Insufficient or arbitrary diversity with InfoGAIL is a direct result of the characteristics of the strategy decoder, $q$, which assigns latent vectors $z$ to various state-action pairs $(s, a)$. The decoder $q$ is modeled by a neural network (NN) and trained without supervision to optimize for MI. It is unconstrained in its assignments. As a result, assignments can vary rapidly within a small region [24, 20]. Fig. 1b (bottom left) visualizes such a decoder. Related behaviors with close-by states could be mapped to unrelated far away regions in the latent space without any meaningful structure. Such assignments can directly influence the characteristics of diversity. Thus, the use of an unconstrained $q$ is not conducive to generalization over latent factors. Additional discussion is in Appendix B.2.

**Prior methods produce misaligned behaviors:** As the strategy decoder influences the diversity in learned behaviors, its regularization is required for generalization. In unsupervised RL, Park et al. [20] impose Lipschitz constraints by the use of Spectral Normalization (SN). It constrains the assigned latent vectors by the Euclidean distance between the corresponding states, scaled by $\lambda_C$, i.e., $||\mu_{q(\cdot|s)} - \mu_{q(\cdot|s')}|| \leq \lambda_C \cdot ||s - s'||$. Fig. 1c (bottom) shows the effect of SN in PointMaze. Assignments are smooth and behaviors are encouraged to deviate from demonstrations uniformly. Such a diversity objective while useful, may disregard the task to be accomplished. *For the purpose of generalization, it is important for diversity to encourage behaviors relevant to the underlying task.*

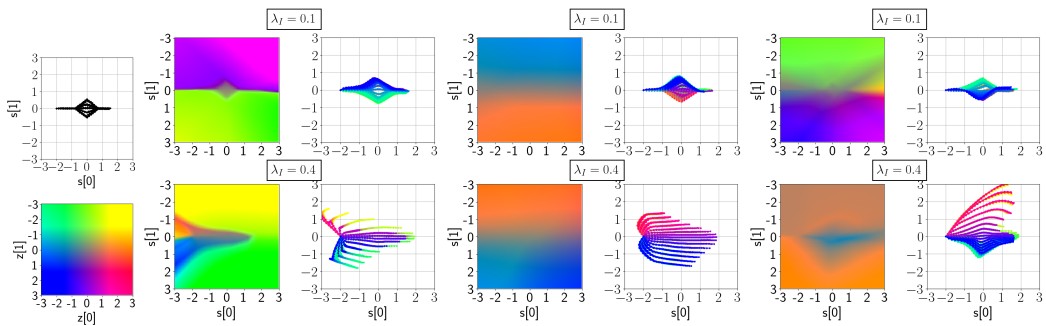

(a) Demos, $Z$  (b) No Regularization (NR) (c) Spectral Normalization (SN)     (d) GSD (Ours)

Figure 1: Fig.1a (Top): The figure visualizes demonstrations for PointMaze. The agent starts at (-2, 0), moves to (2, 0), passing through (-1, 0), (0, $\omega$) and (1, 0) where $\omega \in [-1, 1]$. Fig.1a (Bottom): The figure visualizes the map indicating colors assigned to the 2D latent space. Figs.1b,1c,1d: The figures visualize policy behaviors and latent vectors assigned to the state space under two different importance weights $\lambda_I$. Assignments are visualized by their corresponding colors per Fig. 1a (bottom). We use a state-only decoder for visualizability. Trajectories are colored according to the conditioning latent vectors. Imitation can be easily achieved with a low $\lambda_I$. With high $\lambda_I$, deviation from demonstrations for NR is arbitrary. In addition, the resulting assignment is not smooth, changing rapidly in the neighborhood around the point (-2, 0). With SN, deviations from demonstrations are uniform. With GSD, deviations are contained to the expert distribution in comparison.

## 4  Our Method: Guided Strategy Discovery

In this section, we present our approach, Guided Strategy Discovery (GSD) for generalizable IL from diverse demonstrations. Alternate notions of generalization and further details are in Appendix A.

To encourage InfoGAIL to produce a generalizable latent space, a diversity objective aligned with the underlying task is required. To promote such *task-relevant diversity* we require a task-relevance metric to modulate regularization. Our insight suggests that the discriminator $D$, which captures expert occupancy determined from demonstrations, serves as a valid proxy for task-relevance. We reformulate $D$ to obtain a bounded function $f: S \times A \to (0, 1)$ as shown in Eq. 1, where $\sigma$ is the logistic function, $\lambda_S$, a scaling constant, and $b$, a learnable bias. We then use $f$ to locally modulate Lipschitz constraints as shown in Eq. 2, allowing for dense assignment of latent vectors in high expert-density regions and sparse assignment otherwise. We hypothesize that our regularization encourages diversity in regions of high task-relevance, avoiding a misaligned diversity objective.

$$D(s, a) = \sigma(\lambda_S \cdot f(s, a) + b) \tag{1}$$

$$||\mu_{q(\cdot|s,a)} - \mu_{q(\cdot|s',a')}|| \leq \lambda_C \cdot f(s, a) \cdot ||s - s'|| \tag{2}$$

The constraints are enforced approximately using Lagrange multipliers for the collected on-policy samples. More details are in Appendix C. Fig.1d (bottom) shows the effect of GSD in PointMaze. Learned behaviors make progress towards the goal indicated by demonstrations more consistently.

## 5  Evaluation

To evaluate generalization, we construct an expert policy that accomplishes a task in multiple ways, as determined by a continuous ground truth factor (GTF). We then partition the GTF space into disjoint train and test regions. Demonstrations in $\mathscr{D}$ are generated from the expert policy conditioned on points in the train region. We investigate whether the latent spaces learned by various methods contain behaviors that have similar characteristics as expert behaviors sampled from desired regions in the GTF space. For the sake of evaluation, we consider GTFs that can be directly measured from sampled trajectories and evaluate recovery performance, i.e., if the learned latent space can represent particular expert behavior, in terms of error between desired and measured GTF values. Evaluation for the train region measures in-distribution generalization, as the demonstrations are generated from this distribution. Evaluation for the disjoint test region measures out-of-distribution generalization.

We compare InfoGAIL (IG), InfoGAIL with the reformulated discriminator (Eq. 1) (IG+M), to independently evaluate the modification, IG+M with SN (Sec. 3) (IG+M+SN), to evaluate uniform decoder regularization, and IG+M with our proposed regularization (GSD). As test domains,

we choose InvertedPendulum (IP), HalfCheetah (HC), and FetchPickPlace (FP) [25] with 1D GTFs. In IP, the expert balances the pendulum upright at various slider locations. In HC, the robot is moved at various velocities. In FP, the object is placed at various locations along a line. We consider fifteen demonstrations per domain. Additional details are in Appendix D.1, D.2.

We consider $K$ i.i.d. samples from $p_z(\cdot)$ to search for desired behaviors. We perform limited sampling to reflect the ease of finding these behaviors. We rollout policies conditioned on sampled $z$s and measure the GTFs. We report the least mean absolute error (MAE) between the desired and measured values. We average errors over train and test regions for different $K$, over $1500/K$ rounds, with mean and standard errors reported over five train seeds. We measure task performance of the error-minimizing behaviors over the GTF space with returns.

Recovery performance is shown in Fig. 2. Compared to IG, IG+M performs better with train regions and worse with test regions for IP and HC. The trend reverses with FP. The effect of reformulating the discriminator is unclear and warrants further exploration. IG+M+SN improves over IG with train regions across all domains.

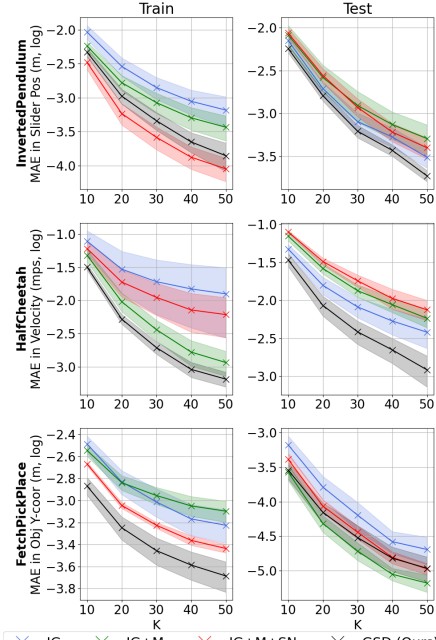

Figure 2: The figure shows recovery errors of four algorithms for the three domains. Shaded regions are standard errors over five train seeds. GSD outperforms baselines for four out of the six considered domain-splits.

However, it improves with test regions only for FP. SN seems better suited to FP as compared to IG, alluding to its hypothesized domain-specific nature. GSD outperforms IG in all cases and other methods in four out of six domain-splits.

Task performance is shown in Fig. 2. GSD outperforms baselines in domains other than HC. The HC robot is rewarded for moving forward with each step. As this reward is not independent of the GT factor (velocity), we consider it uninformative in this domain. Downward trends with increasing K are likely

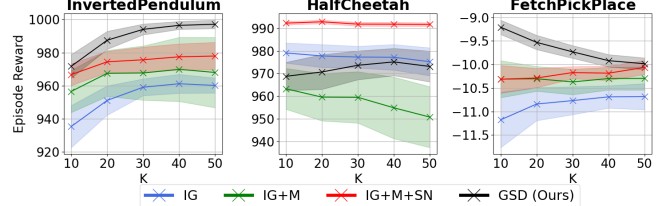

Figure 3: The figure shows the task performance of four algorithms for the three domains. Shaded regions are standard errors over five train seeds. GSD outperforms baselines for two of the three domains.

a result of the error-minimizing behavior compromising reward. Results pertaining to worst-case performance over the GTF space (see Appendix D.3) follow similar trends as average performance. On the whole, GSD recovers GTFs without compromising task performance and reduces recovery error over baselines by 11% with test regions (for $K$=50), contributing a regularization approach appropriate for generalization. We note that latent space analysis is prone to high variance and aim to develop better evaluation frameworks in future work.

## 6    Discussion, Limitations and Future Work

Our work investigates diversity formulations used for learning behaviors from multimodal demonstrations, identifies shortcomings, and proposes a novel regularization method. We evaluate methods for latent factor generalization over three domains and show improved performance.

While we motivate our work to capture latent human preferences, our evaluation is limited to manually curated demonstrations. We aim to collect human demonstrations and conduct user studies to subjectively evaluate learned latent spaces. In our work, the task relevance metric is derived from training demonstrations. To truly generalize to novel behaviors, we aim to investigate latent-conditioned discriminators in future work. Additional discussion is in Appendix E.

**Acknowledgments**

We thank Manisha Natarajan for feedback on the writing. This work was supported by NSF grant CPS-2219755, ONR grant N00014-22-1-2834, MIT Lincoln Laboratory grant FA8702-15-D-0001, and a grant from Ford Motor Company.

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

## A    Related Works

**Generalization**    Generalization or out-of-distribution-ness in RL and IL has been defined in multiple ways. Prior works address behavior learning for tasks or goals that may be specified using simple [75] or expressive descriptors such as language [57, 71, 72, 73, 82] or images [74]. Other works [29, 31, 77, 78, 79] instead focus on behavior learning under varying environment characteristics with a fixed task specification. Both groups study generalization in terms of task accomplishment under varying task specifications or environment characteristics. Our work considers the IL setting with multimodal demonstrators where a task specification is fixed. We thus study generalization from the perspective of the demonstrator, instead of the task or environment. Prior works have addressed behavior learning from demonstrators with differing latent characteristics such as dynamics [69], suboptimality [33, 80], and strategies [15, 53]. However, generalization for behavior learning has rarely been defined or studied in this context. Our work considers generalization in terms of the ability to learn behaviors corresponding to novel latent strategies of the demonstrators.

**Reinforcement Learning**    Early works in RL addressing generalization to unseen dynamics or environments [77, 83] learn policies without explicitly capturing the various behaviors that generalize to novel scenarios. Our work is more related to competence-based unsupervised RL [32, 28, 58, 20, 21] that learn latent spaces to represent various behaviors. However, such methods focus on learning low-level "skills" that can used to perform complex tasks and are not directly evaluated for a particular task or generalization. A related group of works focuses on learning diverse behaviors for a specified task [27, 31, 30] by utilizing rewards alongside unsupervised diversity objectives. Our work is related to learning diverse solutions for novel scenarios in the IL setting where instead of a reward, we have access to a collection of demonstrations.

**Imitation Learning**    IL consists of several works that utilize state distribution matching using forms of divergence [54, 55, 56]. However, they do not directly suit the purpose of multimodal IL with unobserved descriptors. Multimodal IL [18, 19, 33, 51, 69] address captures unobserved factors of demonstrators in a latent space. Tangkaratt et al. [33] and Qiu et al. [69] specifically address demonstrators with diverse qualities and dynamics respectively, and are not suitable for diverse strategies. Wang et al. [51] adopts a VAE pretraining framework where a decoder network is trained to identify latent vectors before performing online imitation. The two-stage process may limit the expressivity of the latent space as it is uninformed of the environment dynamics. Our approach considers learning the decoder network in parallel with behaviors. Our work is similar to Li et al. [18], Wang et al. [51], except that we additionally address their drawbacks towards generalization.

## B    Miscellaneous

### B.1    PointMaze

The PointMaze domain considered in experiments presented in Fig. 1 is a two-dimensional navigation environment with continuous state and action spaces. The state of the environment lies in $[-\infty, \infty]^2$ and represents the x- and y- coordinates of the agent's current location. The action space is a velocity command, which is a two-dimensional vector in $[-1, 1]^2$. The episode length is fixed to 25 steps.

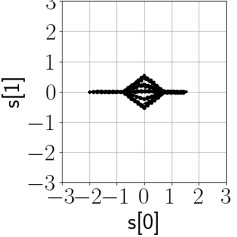

Figure 4:  The figure visualizes expert demonstrations considered for learning behaviors in PointMaze.

### B.2    Unconstrained Decoders

Prior work has found that methods that utilize MI (including InfoGAIL) can be unstable in domains with high dimensional state spaces [27] or involve careful balancing of the objectives [31, 30, 33]. We hypothesize that an unconstrained decoder results in a loss landscape with several local optima

**Algorithm 1** Guided Strategy Discovery

---

**Input**: $\mathscr{D} = \{\tau_i^\xi\}$
**Output**: $\pi$
 1: Initialize policy $\pi$, function $f$, decoder function $q$
 2: Initialize lagrange multiplier $\lambda$, learnable bias $b$
 3: **for** $i \in \{0, 1, 2, ...\}$ epoch **do**
 4:     Sample $z^\pi$ from prior and $\tau^\pi$ using policy $\pi(\cdot|\cdot, z^\pi)$
 5:     Sample $\tau^\xi$ from $\mathscr{D}$
 6:     Define objective for function $f$ and bias $b$: $\forall \chi := (s, a, s') \in \tau$
         $D(\chi) = \sigma(\lambda_S \cdot f(s, a) + b)$
         $J^\mathrm{I} \leftarrow \mathbb{E}_{\tau^\pi}[\log D(\chi^\pi)] + E_{\tau^\xi}[\log(1 - D(\chi^\xi))]$
 7:     Update function f using SGD
         $\theta_r := \theta_r - \eta_r \nabla_{\theta_r} J^\mathrm{I}$
 8:     Define objective for decoder $q$: $\forall \chi := (s, a, s') \in \tau$
         $\delta(\chi) \leftarrow \lambda_C \cdot f(s, a) \cdot ||s - s'|| - ||\mu_{q(\cdot|s,a)} - \mu_{q(\cdot|s',a')}||$
         $q_L(\chi, z) \leftarrow \mathcal{N}(z | \mu_{q(\cdot|s,a)}, \Sigma_{q(\cdot|s,a)})$
         $J^\mathrm{E} \leftarrow \mathbb{E}_{\tau^\pi}[\log q_L(\chi, z^\pi) + \lambda \cdot \min(\delta(\chi), \epsilon)]$
 9:     Update decoder function and $\lambda$ using dual descent
         $[\theta_\phi, \lambda] := [\theta_\phi, \lambda] + [\eta_\phi \nabla_{\theta_\phi} J^\mathrm{E}, -\eta_\lambda \nabla_\lambda J^\mathrm{E}]$
10:     Update policy $\pi$ with PPO using rewards
         $r(s, a, s', z^\pi) = (1 - \lambda_\mathrm{I}) \cdot -\log(-f(s, a)) + \lambda_\mathrm{I} \cdot \log q_L(\chi, z^\pi)$
11: **end for**
12: **return** $\pi$

---

and makes the optimization unstable. This is true especially in high dimensional domains where it is easy to overfit to specific states and maximize MI, but not any diversity that is meaningful.

## C  Algorithm

The algorithm is shown in Alg. 1. At equilibrium of the adversarial optimization in GAIL, the discriminator $D$, and by extension $f$, output a constant value across the state-action space [65]. However, $f$ still captures expert occupancy during policy optimization during which it guides decoder regularization.

We use the public code-base for VILD [33] as our starting point: github.com/voot-t/vild_code. To ensure that the discriminator, which guides the regularization is smooth, we use gradient penalty [66] with a weight of 0.2. We perform spectral normalization [60] using `torch.nn.utils.parametrizations.spectral_norm`. To implement Lipschitz constraint scaling with $\lambda_S$, we scale the inputs to the decoder.

## D  Evaluation

### D.1  Framework, Domains, Datasets

To further simplify our evaluation process, we consider the 1D ground truth latent space to be divided into five equal-sized intervals. We consider the three non-consecutive intervals to represent the train region and the remaining two to the test region. For generating expert demonstrations, we choose the mean value in each interval, add small Gaussian noise, and then condition the expert policy. For errors measured in recovery performance, we choose the mean ground truth factor value of each interval to represent the entire interval. Thus train and test errors are averaged over three and two ground truth latent factor values respectively.

In InvertedPendulum, the default reward function is used from the environment, which awards +1 for each step the pendulum is upright within a few degrees with the upward normal. For HalfCheetah, the reward is +1 for each time step if the cheetah moves forward by a non-zero amount. For Fetch, the

task is to move the object from its initial position to a particular x-coordinate (the y-coordinate does not matter). The reward is measured as the sum of absolute differences between the current and target x-coordinates. InvertedPendulum and Fetch are deterministic environments, while HalfCheetah is a stochastic environment. The episodes for InvertedPendulum, HalfCheetah, and FetchPickPlace are of fixed lengths of 1000, 1000, and 200 steps respectively.

Demonstrations for InvertedPendulum staying upright at various mean positions along the slider in the range [-1, 1] m. Demonstrations for HalfCheetah consist of the robot running at different mean velocities [1, 2, 3, 4, 5] m/s. Demonstrations for Fetch consist of the robot arm picking the object up from the initial location and placing it at varying y-coordinates at the target x-coordinate where y lies in the range 0.75 + [-0.2, 0.2] m. We consider five demos for each interval in the train region resulting in a training dataset with 15 demonstrations per domain.

## D.2 Training Details

With Fetch, vanilla InfoGAIL did not converge to a proper policy with our demonstration dataset. To ensure convergence of the baseline, we used observation normalization with running statistics and augmented PPO loss with a decaying behavior cloning (BC) loss [67]. For input $z$ for the BC loss, we used the mean of $z$s (weighted by variance) inferred from the latest strategy decoder $q$.

Each method is independently tuned for $\lambda_I$ (and $\lambda_C$ for SN, GSD) to maximize MAE for K=10 over averaged over four rounds of evaluation and five train seeds. We pick the optimal $\lambda_I$ from [0.1, 0.2, 0.3] and $\lambda_C$ from [0.5, 1.0, 5.0, 10.0] (InvertedPendulum), [0.02, 0.05, 0.1, 0.2] (HalfCheetah) and [0.1, 0.5, 1.0, 5.0] (FetchPickPlace).

## D.3 Worst Case Analysis

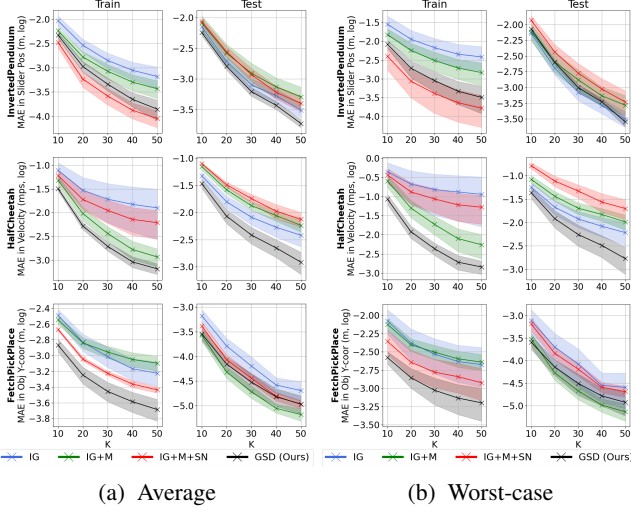

(a) Average   (b) Worst-case

Figure 5: The figures show recovery errors of four algorithms for the three domains. Shaded regions are standard errors over five train seeds. Fig. 5a (copy of Fig. 2) and Fig. 5b show the recovery errors on an average- and worst-case basis across GT factors in a particular region (train/test) respectively. The trends remain largely consistent across the two evaluation modes showing GSD outperforming the other methods in four of the six considered domain-splits.

In Sec. 5, we present results for recovery and task performance averaged over all GT factors in the train and test regions. While such results are useful in understanding if GT factors are recovered well on average, the performance for particular factors might vary significantly from the mean. Understanding the worst-case recovery performance across GT factors helps bound the performance across the factor space. To measure worst-case recovery performance, we first compute the recovery performance for a particular GT factor across evaluation rounds and train seeds. We then report

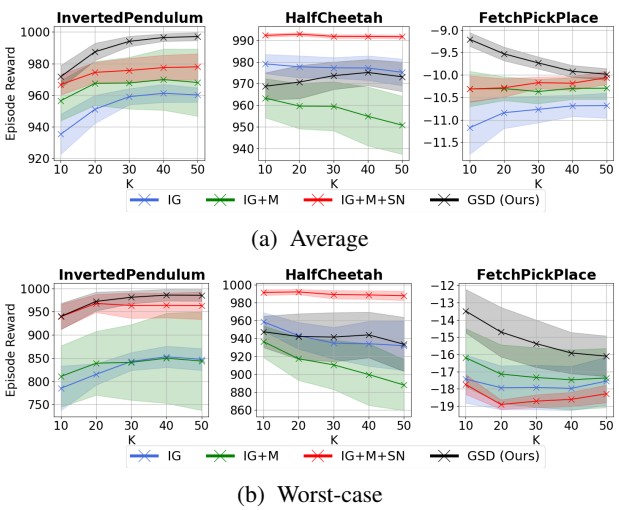

(a) Average

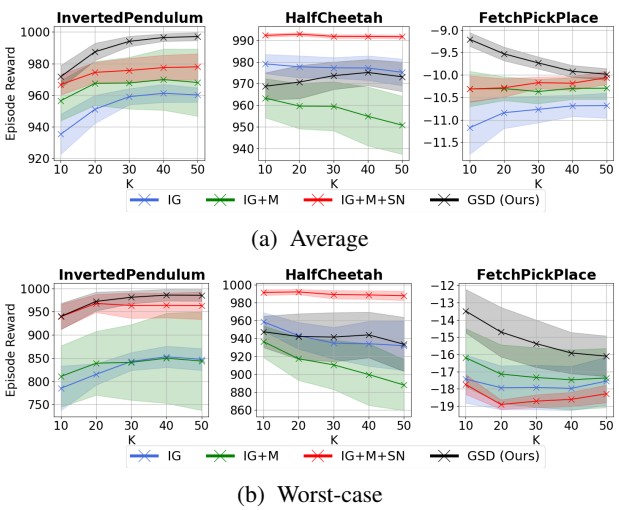

(b) Worst-case

Figure 6: The figure shows the task performance of four algorithms for the three domains. Shaded regions are standard errors over five train seeds. Fig. 6a (copy of Fig. 3) and Fig. 6b show the task performance errors on an average- and worst-case basis across all GT factor values respectively. GSD outperforms baselines for two of the three domains for both evaluation models.

the performance for the GT factor that has the least MAE (for K=50) among factor values in the train and test regions separately. Similarly, for worst-case task performance, we average the task performance of MAE-minimizing behaviors across evaluation rounds and train seeds for each GT factor value. We then report the worst task performance across all factor values. For both metrics, we report standard errors computed over train seeds.

Worst-case recovery performance alongside average recovery performance is shown in Fig. 5. Trends across methods for worst-case performance remain largely consistent with average performance, except for the test regions of InvertedPendulum and FetchPickPlace. In InvertedPendulum, GSD is comparable with IG worst-case but outperforms the other methods on average. In FetchPickPlace, GSD is comparable with SN on average but outperforms SN and IG in worst-case performance. Worst-case task performance alongside average task performance is shown in Fig. 6. GSD outperforms the other methods for InvertedPendulum and FetchPickPlace on average and worst-case. Notably, the task performance of SN decreases considerably for FetchPickPlace. This provides evidence that SN may be sacrificing task completion when producing novel behaviors. We leave further qualitative analysis for future work.

# E    Limitations, Future Work

Our work pertains to the limited realm of adversarial IL frameworks that employ diversity objectives in the form of MI. The generalization capabilities of other multimodal IL frameworks based on non-adversarial IL should be explored.

In our method, the notion of task relevance for guiding regularization is captured by the expert discriminator learned in the GAIL framework. Alternate forms (e.g., disentangled rewards [51]) could be explored, as they offer regularization over other aspects of the task and environment. Yet, such discriminators or reward functions are fundamentally limited to the training demonstrations. Approaches that learn more generalizable reward functions [84] could be considered in conjunction with our regularization.

The scope of generalization considered in this work pertains to variations in the demonstrated expert strategies. Other forms of generalization to altered environment dynamics, adversarial perturbations, etc., should also be considered in the context of imitation learning.

