# OpenReview forum: "Generalized Behavior Learning from Diverse Demonstrations"
_robot-learning.org/CoRL/2023/Workshop/OOD — OOD Workshop @ CoRL 2023_

### Official Review · Reviewer_FkhG · 2023-10-15
**Good, well presented technique, but no explicitly treatment of OOD**

**Rating:** 7
**Confidence:** 4

**Review:**

Title: Generalized Behavior Learning from Diverse Demonstrations

Submission Guidelines followed: Yes!

The paper proposes a new objective term in the standard imitation learning loss (specifically the GAIL formulation). The new objective term is formulated to encourage latent space vector occupancy in latent space regions where expert demonstrations are abundant. This, combined with varying expert demonstrations, is done to capture the diversity in the expert demonstrations better.

The paper presents strong experimental results showing that their method performs better in imitation learning expert behavior recovery than competing approaches.

The main weaknesses of this paper, as related to out-of-distribution (ODD) analysis are:
- the paper does not discuss how expert behaviors can be generated, it is assumed that for this imitation learning technique to work out-of-distribution, the out-of-distribution expert demonstrations must be available; as far as I can tell this is a improved latent space interpolation technique, which does not necessarily generalize outside of the training set
- the results presented in the paper (at least so far) are average instead of worst-case; the argument for the regularization technique is that it can capture diverse expert behaviors and so, likely, a better metric to report would be worst-case recovery loss across the domain span

General comments:
- Quality of Writing: High
- Novelty of Approach: Medium (the approach is novel)
- Significance of the Problem: Medium (existing methods appear to work well)
- Technical Strengths: Clear explanation of contributions with illustrative results
- Technical Weakness: The approach does not appear to treat OOD explicitly
- Quality of Results: High, the results and visualization are done very well

---

### Official Review · Reviewer_WyFe · 2023-10-16
**Weak accept because tangentially related**

**Rating:** 7
**Confidence:** 4

**Review:**

This paper focuses on augmenting imitation learning with the ability to learn a meaningful representation of diversity in the expert demonstrations. They discuss existing methods of learning mutual information between tasks and the impact of regularization using Spectral Normalization. They identify shortcomings in these methods, and develop a new regularization method that promotes task-relevant diversity and outperforms the previous benchmarks. The method is tested on various OpenAI Gym problems with variations in task distribution.

This paper is well-written, builds significantly upon prior work, is technically sound, and demonstrates convincing results.

However, the relevance of the paper to the OOD workshop is debatable. The paper demonstrates a method of generalizing to different types of expert demonstrations. However, it's not clear that this constitutes different distributions. For instance, they test on InvertedPendulum where the expert demos balance the pendulum at different positions, which are all arguably distinct tasks drawn from the same distribution. However, this paper does highlight an important problem of training for generalizability, and promoting a representation of diversity of tasks.

I classify this paper as tangentially related to the OOD workshop, and therefore a weak reject.

---

### Decision · Program_Chairs · 2023-10-17

**Decision:**

Accept

**Comment:**

We agree with the reviewers’ assessment that this work is technically sound and will contribute to productive, topical discussions at the 2023 Workshop on OOD Generalization in Robotics. In particular, we appreciate that this paper demonstrates convincing policy learning results, but would like to stress the reviewers' comments that the impact of your work (in the context of this workshop) would be improved by clarifying the specific distributions involved in OOD generalization. We recommend the authors incorporate the reviewers’ feedback into their camera-ready submission to further improve their manuscript.